# Pathogenesis and defense mechanism while *Beauveria bassiana* JEF-410 infects poultry red mite, *Dermanyssus gallinae*

**So Eun Park**[1], **Jong-Cheol Kim**[1], **Yeram Im**[1], **Jae Su Kim**[1,2]*

**1** Department of Agricultural Biology, College of Agriculture & Life Sciences, Jeonbuk National University, Jeonju, Korea, **2** Department of Agricultural Convergence Technology, Jeonbuk National University, Jeonju, Korea

* jskim10@jbnu.ac.kr

**Data Availability Statement:** All relevant data are within the manuscript and its Supporting information files.

## Abstract

The poultry red mite, *Dermanyssus gallinae* (Mesostigmata: Dermanyssidae), is a major pest that causes great damage to chicken egg production. In one of our previous studies, the management of red mites using entomopathogenic fungi was evaluated, and the acaricidal fungus *Beauveria bassiana* JEF-410 was selected for further research. In this study, we tried to elucidate the pathogenesis of *B. bassiana* JEF-410 and the defense mechanisms of red mites at a transcriptome level. Red mites collected from a chicken farm were treated with *B. bassiana* JEF-410. When the mortality of infected red mites reached 50%, transcriptome analyses were performed to determine the interaction between *B. bassiana* JEF-410 and red mites. Uninfected red mites and non-infecting fungus served as controls. In *B. bassiana* JEF-410, up-regulated gene expression was observed in tryptophan metabolism and secondary metabolite biosynthesis pathways. Genes related to acetyl-CoA synthesis were up-regulated in tryptophan metabolism, suggesting that energy metabolism and stress management were strongly activated. Secondary metabolites associated with fungal up-regulated DEGs were related to the production of substances toxic to insects such as beauvericin and beauveriolide, efflux pump of metabolites, energy production, and resistance to stress. In red mites, physical and immune responses that strengthen the cuticle against fungal infection were highly up-regulated. From these gene expression analyses, we identified essential factors for fungal infection and subsequent defenses of red mites. These results will serve as a strong platform for explaining the interaction between *B. bassiana* JEF-410 and red mites in the stage of active infection.

## Introduction

The poultry red mite *Dermanyssus gallinae* (Mesostigmata: Dermanyssidae) is a serious pest that reduces egg yields and reduces chicken production worldwide [1, 2]. A hematophagous ectoparasite, the poultry red mite requires energy from blood for development and oviposition. Although they mainly feed on the blood of birds, they can infest mammals including

**Funding:** This work was supported by the National Research Foundation of Korea(NRF) grant funded by the Korea government(MSIT) (NRF-2018R1A2B6001351). The funder had no role in study design, data collection and analysis, decision to publish, and preparation of the manuscript.

**Competing interests:** The authors have declared that no competing interests exist.

humans [3, 4] and can transfer zoonotic diseases such as fowl spirochaetosis, chicken pox virus, Newcastle virus, pullorum disease, fowl typhoid, and fowl cholera [5]. The reproduction rate of poultry red mites is very fast, with a pair of mites reproducing to 8 mites after 1 week, 64 after 2 weeks, 32,768 after 5 weeks, and 1 billion after 10 weeks [6]. Thus, careful management tools are important early in occurrence.

It has been a common practice to use chemical acaricides containing amitraz, carbaryl, and permethrin to control red mites. However, repeated use of acaricides has resulted in a high degree of resistance of red mites to the chemicals [7, 8]. In addition, chemicals that accumulate in the organs and tissues of livestock are a threat to the environment as well as to humans [9]. Another factor is that the chemical acaricides are applied by direct spraying on chickens, releasing them into the environment as toxins. Therefore, an eco-friendly control method for application to chickens or eggs should be developed.

The temperature and humidity of poultry farms that the red mites inhabit are suitable for survival, growth, and colonization of entomopathogenic fungi. It has been reported that the entomopathogenic fungi *Metarhizium anisopliae* and *Beauveria bassiana* are effective in controlling mites in chickens [10–12]. We also confirmed that *B. bassiana* JEF-410 has high acaricidal activity in laboratory and field conditions [13]. However, the activity of entomopathogenic fungi is slower than that of chemical acaricides. This might be due to red mite defense responses, including immune mechanisms, although environmental factors also could be of influence. Therefore, to control red mites more effectively using entomopathogenic fungi, we need to determine the response mechanism of red mites to the fungal infection and that of the fungus to the mites' defenses.

From a review of transcriptome analysis using the *Isaria fumosorosea* infected transcriptome of *Plutella xylostella* was analyzed. As a result, immune recognition families such as toll and Imd pathway, melanization, and antimicrobial peptides (AMPs) were inhibited over time. This suggests the potential of fungi as biopesticides [14]. In that work, the possibility of insecticidal activity of entomopathogenic fungi can be confirmed by identifying up-regulated genes. Other reports interpreted the resistance mechanism in mites using the results of according to pathogens or external attacks [15, 16]. When considering the use of entomopathogenic fungi for control of target insects, it is necessary to understand the interaction between fungi and the host insect at the transcriptional level. Such information on pathogenicity and immune-related genes will be helpful in regulating the activity of virulence-related genes. However, molecular mechanisms involved in mite resistance to acaricides or microorganisms are poorly understood [17, 18].

In this study, we analyzed the expression levels of RNAs during JEF-410 infection of red mites. The total RNA of red mites in the active infection stage was extracted, as was that from non-infecting *B. bassiana* JEF-410 and non-infected red mites as controls. Transcriptome analyses were conducted to investigate the interactions between red mites and the fungal pathogen during infection. Using the raw sequencing data, we performed analyses of differently expressed genes (DEG), gene ontology (GO), gene ontology enrichment (GO enrichment), and the KEGG pathway to identify genes related to the pathogenicity of *B. bassiana* JEF-410 and genes related to the red mite response. Analysis of the function of genes expressed in *B. bassiana* JEF-410 and red mites elucidated the interactions involved in infection, and these data could be used to improve the acaricidal activity of *B. bassiana* JEF-410.

## Materials and methods

### Fungal isolate of *Beauveria bassiana* JEF-410

Entomopathogenic fungus *B. bassiana* isolate JEF-410 was collected from mountain soil in Korea (37˚16'10.4"N 128˚46'48.9"E, Jeongseon, Kangwon-Do, S. Korea) using an insect baiting

method [19], and the genomic DNA of collected fungus was extracted and identified by sequencing internal transcribed spacer sequence (ITS) region [19]. The isolate was cultured on quarter-strength Sabouraud dextrose agar (¼ SDA, Difco, USA) and maintained in an incubator at 25˚C for 14 days, and the cultured fungal block was placed in a 20% (v/v) glycerol stock solution (Glycerol, Daejung, Korea) and stored at -80˚C (Insect microbiology and biotechnology laboratory, Jeonbuk National University, Jeonju, Korea) [20].

## Red mite (*Dermanyssus gallinae*)

Red mites were collected from a local chicken farm (35˚20'39.4"N 126˚59'73.7"E, Damyang, Korea). Most collected red mites were colonized in the iron frames of chicken rearing structures. The mite colonies were harvested using a fine brush and placed in a breeding dish (50×15×40 mm$^3$) [21]. The collected mites were moved to a laboratory within two hours. Under dry conditions, natural mortality is relatively high; the collected mites were kept in high humidity conditions. Of the colonies, adults and larvae were removed, and only the nymph stage was used in this experiment.

## Sample preparation and RNA extraction

For RNA transcriptome analysis of infecting *B. bassiana* JEF-410 and infected red mites, non-infecting *B. bassiana* JEF-410 and non-infected red mites served as controls, and *B. bassiana* JEF-410-infected red mites were the main investigation. Each treatment group was sampled in triplicate.

RNAs of red mites infected by *B. bassiana* JEF-410 were extracted from about 500 infected nymphs collected from a chicken farm. All the red mite nymphs were placed on the NC membrane (Amersham™ Protran™ NC membrane, pore size 0.45 μm, Cytiva, England) layered on a 1/4SDA medium plate (90 × 15 mm$^2$) where *B. bassiana* JEF-410 was cultured for 7 days. Two and a half days after infestation, infected red mites were transferred to a 1.5-ml micro tube. To obtain fresh total RNAs, the sample was kept at -70˚C for 24 hours and then freeze-dried for 2 days.

As a control, red mite RNA was extracted from 500 non-infected nymphs. Non-infected red mites were transferred to a 1.5 ml micro tube and freeze-dried under the same conditions as described above. As a non-treated control fungus, *B. bassiana* JEF-410 RNA was extracted from mycelia. Conidia suspension (1×10$^7$ conidia/ml) of *B. bassiana* JEF-410 was spread on the NC membrane covering 1/4SDA medium plate and cultured for days at 25˚C. Only mycelia produced on the NC membrane was harvested and freeze-dried under the same conditions as described above.

The total RNAs of three samples were extracted by TRIzol reagent (Molecular Research Center Inc., Cincinnati, OH, USA) following the manufacturer's instructions. The quality and concentration of total RNA were determined by measuring the absorbance ratio at 260 and 280 nm and electrophoresis bands with 1.5% agarose gel. The integrity of the extracted RNA was measured by an Agilent 2100 bioanalyzer (Agilent Technologies Korea Ltd., Seoul, South Korea).

## cDNA library construction

Sequencing libraries of each RNA sample were made using the Truseq RNA kit (Illumina, San Diego, CA, USA) according to the manufacturer's protocol. The cDNA library construction and Illumina sequencing of the samples were performed (Macrogen Corporation, Seoul, South Korea). Poly-A containing mRNA molecules were purified using poly-T oligo-attached magnetic beads. The treated mRNA was broken into small fragments at elevated temperature

using divalent cations. The fragmented mRNAs were reverse transcribed into the first cDNA using random primers. The next strand of cDNA was synthesized using DNA polymerase I and RNase H. The cDNA fragment underwent a repair step with a single dATP and a process of ligation of the sequencing adapter. Finally, the product was purified and concentrated by PCR to generate a cDNA library. The product was sequenced on an Illumina HiSeq2000 sequencer (Illumina, San Diego, CA, USA) to generate high-throughput transcript sequence data with an average read length of 101 bp.

## Differentially expressed gene analysis

The obtained short read sequences of *B. bassiana* JEF-410, non-treated control mites, and fungal-treated mites were checked for quality using fastQC ver. 0.11.7 [22]. The low-quality reads of samples were trimmed using fastp ver. 0.23.2 (https://www.bioinformatics.babraham.ac.uk/projects/fastqc/). The mite sequences were assembled using Trinity ver. 2.13.0 with the min_contig_length option set to 150 after pooling reads of non-treated and fungal-treated mites. The sequences of the coding region were extracted from the assembled sequences using Transdecoder ver. 5.5.0 (https://github.com/TransDecoder/TransDecoder/releases). In order to quantify transcript abundances of *B. bassiana* JEF-410 and red mites, the transcripts per million (TPM) values were calculated by mapping short reads of *B. bassiana* JEF-410 and fungal-treated red mites to the assembled red mite sequences and short reads of non-infecting fungus *B. bassiana* JEF-410 and infecting fungus to the *B. bassiana* ARSEF8028 (ASM168263v1) sequences using kallisto ver. 0.45.0 [23].

DEG analysis of *B. bassiana* JEF-410 and red mites was performed using tpm and est_lead counts from kallisto. The edgeR package (https://bioconductor.org/packages/devel/bioc/html/edgeR.html.) was used to collect three repeated count values of each sample. As a result, DEG was compared with the confirmed FDR value and $\log_2$FC value. First, a value with a logFDR value of 1.5 or more was determined as a significant analysis value. After that, up- and down-regulated short reads were counted with a value of $|\text{Log2FC}| > 1$.

## Analysis of functional changes in *B. bassiana* JEF-410 and red mite

Gene ontology (GO) analysis was performed with InterProScan function of Blast2GO (https://www.blast2go.com/) based on the EMBL-EBL database by grouping up- and down-regulated DEGs of *B. bassiana* JEF-410 and red mites. The GO analysis results confirmed the change in GO function of DEGs by counting the number of genes annotated in three GO groups (biological process, cellular component, molecular function) based on GO level 3. KEGG pathway analysis was performed on the up- and down-regulated DEGs of *B. bassiana* JEF-410 and red mites by the BBH method in the KEGG Automatic Annotation Server (KAAS, https://www.genome.jp/kegg/kaas/). The DEGs of red mites were classified into two groups based on up- and down-regulated genes ($|\log_2$FC$| > 1$). In order to conduct GO enrichment analysis, the DEGs of red mites were identified at an E-value threshold of $1.0 \times 10^{-10}$ based on the database of *Tetranychus urticae* (ASM23943v1, Ensembl database) using the blastx function of Blast2GO. Arthropoda immune-related genes group, respectively. For the analysis of immune-related genes, annotated in uniport database were identified in the UniProt (https://www.uniprot.org/).

The DEGs of *B. bassiana* JEF-410 were classified into two groups of up- and down-regulated genes ($|\log_2$FC$| > 1$). In order to conduct GO enrichment analysis, the DEGs of *B. bassiana* were identified at an E-value threshold of $1.0 \times 10^{-10}$ based on the database of *Saccharomyces cerevisiae* (Saccharomyces Genome Database, http://sgd-archive.yeastgenome.org/sequence/S288C_reference/orf_dna/). GO enrichment analysis was performed for each gene

group by g:profiler [24] with a Benjamini-Hochberg false discovery rate (FDR) < 0.05. For analysis of metabolic genes, those annotated in the Uniport database were identified in the NCBI nr-database (https://www.ncbi.nlm.nih.gov/).

## Gene expression validation by qRT-PCR

The three different RNA samples (*B. bassiana* JEF-410, non-treated mites, and fungal-infected red mites) were used to synthesize cDNA for RT-PCR and quantitative RT-PCR (qRT-PCR) in three replicates. A 1 ul aliquot of each RNA was subjected to reverse transcription (RT) using AccuPower® RT PreMix (Bioneer, Daejeon, Republic of Korea) with the oligo (dT) 15 primer (Promega, MI, USA). A set of primers (S1 Table) for RT-PCR was designed at Primer3-Plus (https://www.bioinformatics.nl/cgi-bin/primer3plus). RT-PCR was performed as follows: an initial denaturation at 94˚ C for 5 minutes, followed by 34 cycles of 30 seconds at 94˚ C, 30 seconds at 59˚ C, and 30 seconds at 74˚ C, followed by a final extension for 10 minutes at 74˚ C (C-1000, Bio Rad, Hercules, CA, USA). The target genes of the samples were determined by 1.5% agarose gel electrophoresis. qRT-PCR was performed using Thunderbird® Syber® qPCR mix (QPS-201, TOYOBO, Japan) on a 96-well Bio-Rad CFX96 Real-Time PCR System (Bio-Rad, USA). Cycling parameters for qRT-PCR were as follows: denaturation for 1 minute at 95˚ C, 40 cycles of 15 seconds at 95˚ C, and 1 minute at 60˚ C, followed by melting with an increase in temperature of 0.5˚ C per 5 seconds from 65˚ C to 95˚ C. The *B. bassiana* actin (= γ-actin, GenBank Accession No: HQ232398) and red mite cytochrome p450 (GenBank Accession No: MN695339.1) were used as internal controls. ΔCt (threshold cycle) was calculated as (Ct value of up-regulated genes)—(Ct value of *Ma*-actin) and subjected to the calculation of fold change value ($2^{-\Delta\Delta Ct}$). All experiments were performed in three replicates. Statistical analyses were performed using Student's t-test, and a p-value < 0.005 was considered to indicate a significant difference.

## Results

### Analysis of raw sequence data from infecting fungus and infected mites

The gene expression levels of *B. bassiana* JEF-410 and red mites were analyzed using the RNAs of *B. bassiana* JEF-410-infected red mites, and transcriptional changes occurring during the infection in red mites were predicted. For this analysis, non-infecting fungus and non-infected red mites served as controls. The analysis time point was fixed to 60 hours after fungal treatment, the $LT_{50}$ of the fungus-treated red mites [13]. This could just describe the up- and down-regulation in the determined time point. The reason for setting this time point was that the pathogenesis could be proceeding, the pathogenesis-related genes would be more highly up-regulated and finally back to normal expression when the fungal infection and propagation is over.

Total RNA was extracted from fungus, fungus-infected red mites, and non-infected red mites, and each treatment was replicated three or four times (S2 Table). From non-infecting *B. bassiana* JEF-410, 36,616,478 total and 16,171,238 raw RNA-seq reads were obtained. From the non-infected red mites, a total of 17,648,091, 18,269,132 and 17,347,996 raw RNA-seq reads were obtained. These raw data sequences were used as controls. After fungal treatment, a total of 16,332,684, 12,462,391 and 18,294,405 raw RNA-seq reads were obtained from *B. bassiana* JEF-410-infected red mites. Low-quality sequences were trimmed using a Fastp program. Unlike *B. bassiana*, *de novo* assembly was performed to obtain contigs for analysis of red mite gene expression levels because there is no reference sequence available. The total number of assembled red mite contigs was 82,621 bp, and the $N_{50}$ value was 1,260 bp. Among them, the

number of contigs corresponding to the coding region was 16,621, and the $N_{50}$ value was 1,791 (S3 Table).

## Differentially expressed genes (DEGs) in infecting fungus and infected mites

The short reads of fungus-treated mites and *B. bassiana* JEF-410 were mapped to *B. bassiana* ARSEF8028 (ASM168263v1) sequences as a reference. Similarly, the short reads of non-treated mites and fungus-treated mites were mapped to the *de novo* assembled red mite sequences. Distribution of each DEG was counted based on the |Log2FC|>1 value. The expression levels of 583 genes were significantly changed in infecting *B. bassiana* JEF-410 (Fig 1A). Up-regulated and down-regulated genes in the *B. bassiana* genome numbered 471 and 112, respectively. Expression levels of 1,639 genes were significantly changed in the red mites treated with *B. bassiana* JEF-410 (Fig 1B). Up- and down-regulated genes numbered 1,249 and 390, respectively. Among the genes, those up- or down-regulated by greater than two-fold or with a p-value for cutoff FDR of 0.01 numbered 638 and 120, respectively. The analysis of heatmap was performed to investigate the variation of gene expression among the repetitions in each treatment. The heat map result showed similar patterns of expression in each repetition of the same group.

## Functional classification of *B. bassiana* and red mite DEGs

In *B. bassiana* JEF-410 GO analysis, up-regulated genes were much more frequent than down-regulated genes (Fig 2). In the GO analysis of DEGs, 473 up-regulated genes and 318 down-regulated genes were identified. Of these, 97 up- and 15 down-regulated genes were involved in a biological process, 29 and 7 in a cellular component, and 146 and 27 in a molecular function, respectively. These correspond to respective proportions of 34.9% biological process, 11.2% cellular component, and 53.9% molecular function. The up-regulated genes of biological process were annotated to groups related to metabolic processes such as cellular metabolism, nitrogenous metabolism (GO:0006807), and primary metabolism (GO:0044238); the down-regulated biological process genes were annotated to functions of reproduction (GO:0000003) and the reproductive process (GO:0022414) in multi- and single-celled species. A large number of up-regulated genes in molecular function was annotated to catalytic activity (GO:0003824), binding (GO:0003677), hydrolase activity (GO:0016787), ion binding (GO:0043167), and oxidoreductase activity (GO:0016491). The DEG groups of molecular function showed significant difference, the up-regulated genes were annotated to a cellular component, and down-regulated genes were annotated to only molecular function with a cell pole (GO:0007279), a catalytic complex (GO:1902494), and cell projection (GO:0042995).

In the GO analysis of red mites, similar to the *B. bassiana* JEF-410 GO analysis, up-regulated genes were much more frequent than down-regulated genes; there were 638 up-regulated genes and 120 down-regulated genes (Fig 3). The up- and down-regulated DEGs were annotated to a biological process (86 and 3, respectively), a cellular component (13 and 21), and molecular function (115 and 21). The proportions annotated for each GO term were biological process 32.7%, cellular component 14.3%, and molecular function 53.0%. The most common grouped term was molecular function, and up-regulated genes were annotated to catalytic activity (GO:0003824), binding (GO:0003677), and organic and heterocyclic compound binding (GO:0097159). Many up-regulated genes in the cellular, metabolic, and biosynthetic biological processes were annotated, and down-regulated genes were annotated in response to stress (GO:0006950), establishment of localization (GO:0051649), catabolic process (GO:0009056), cellular response to stimulus (GO:0051716), transmembrane transport

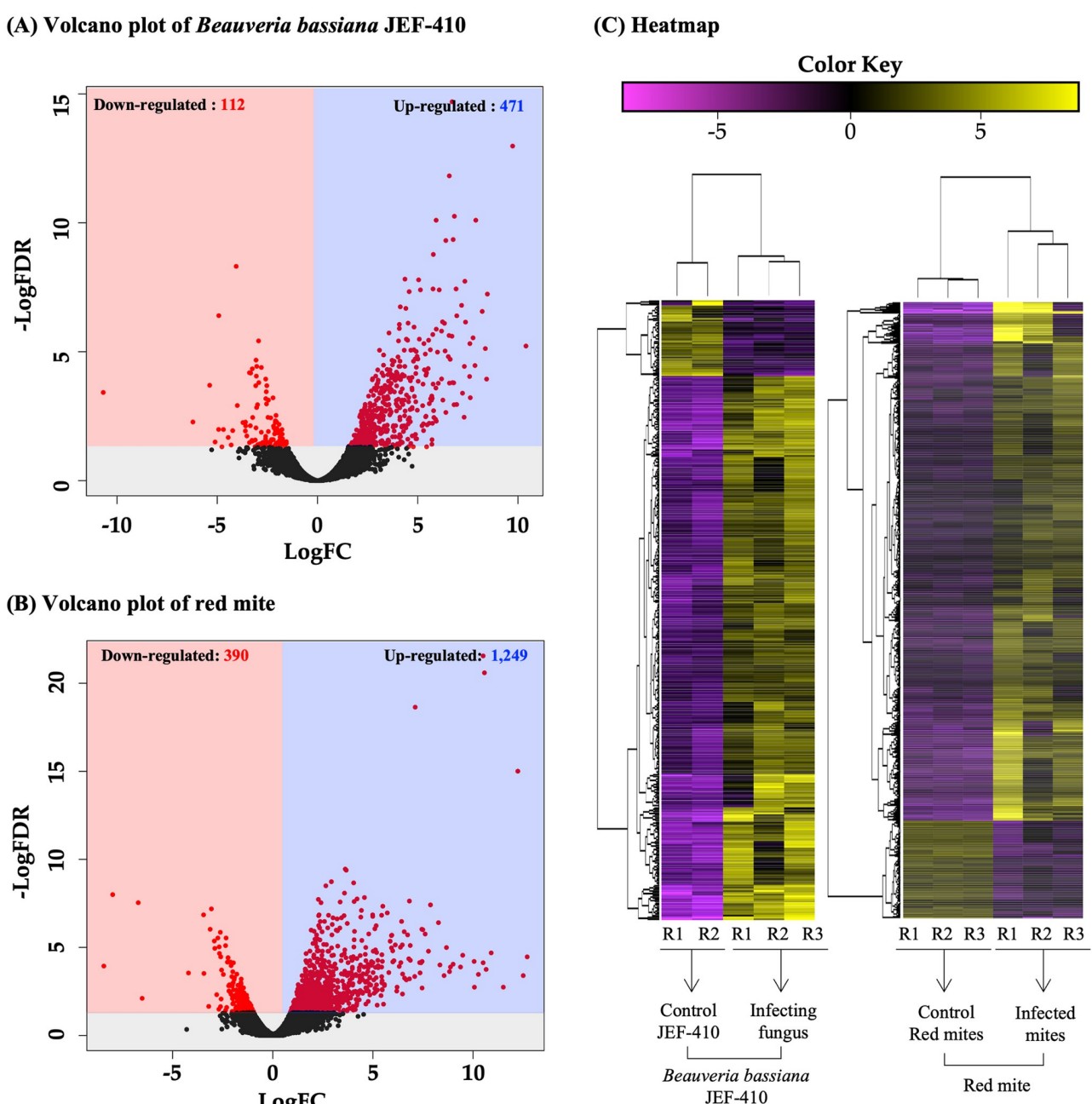

**(A) Volcano plot of *Beauveria bassiana* JEF-410**

**(B) Volcano plot of red mite**

**(C) Heatmap**

**Fig 1. Volcano plot and heatmap of *Beauveria bassiana* JEF-410 and red mite DEGs.** Distribution of DEGs in *Beauveria bassiana* JEF-410 and red mites. (A) Volcano plot comparing log2 ratios of TPM expression values (FDR < 0.05) of non-treated control fungus vs. red mite-infecting fungus. (B) Volcano plot comparing log2 ratios of TPM expression values (FDR < 0.05) of non-treated control red mites vs. fungal-infected red mites (C) Heat map of *B. bassiana* JEF-410 and red mite. The meaning of the 'R' attached to each number is the repetition of the sample.

(GO:0055085), and localization (GO:0051179). The up-regulated genes of the cellular component were annotated to protein-DNA complex (GO:0032993), chromatin (GO:0000790), DNA packaging complex (GO:0000786), and catalytic complex (GO:1902494), and down-regulated genes were annotated to the prefoldin complex (GO:0016272). There was a significant difference between DEG groups in molecular function. The up-regulated genes were annotated to GO terms related to catalytic activity (GO:0003824), binding (GO:0003677),

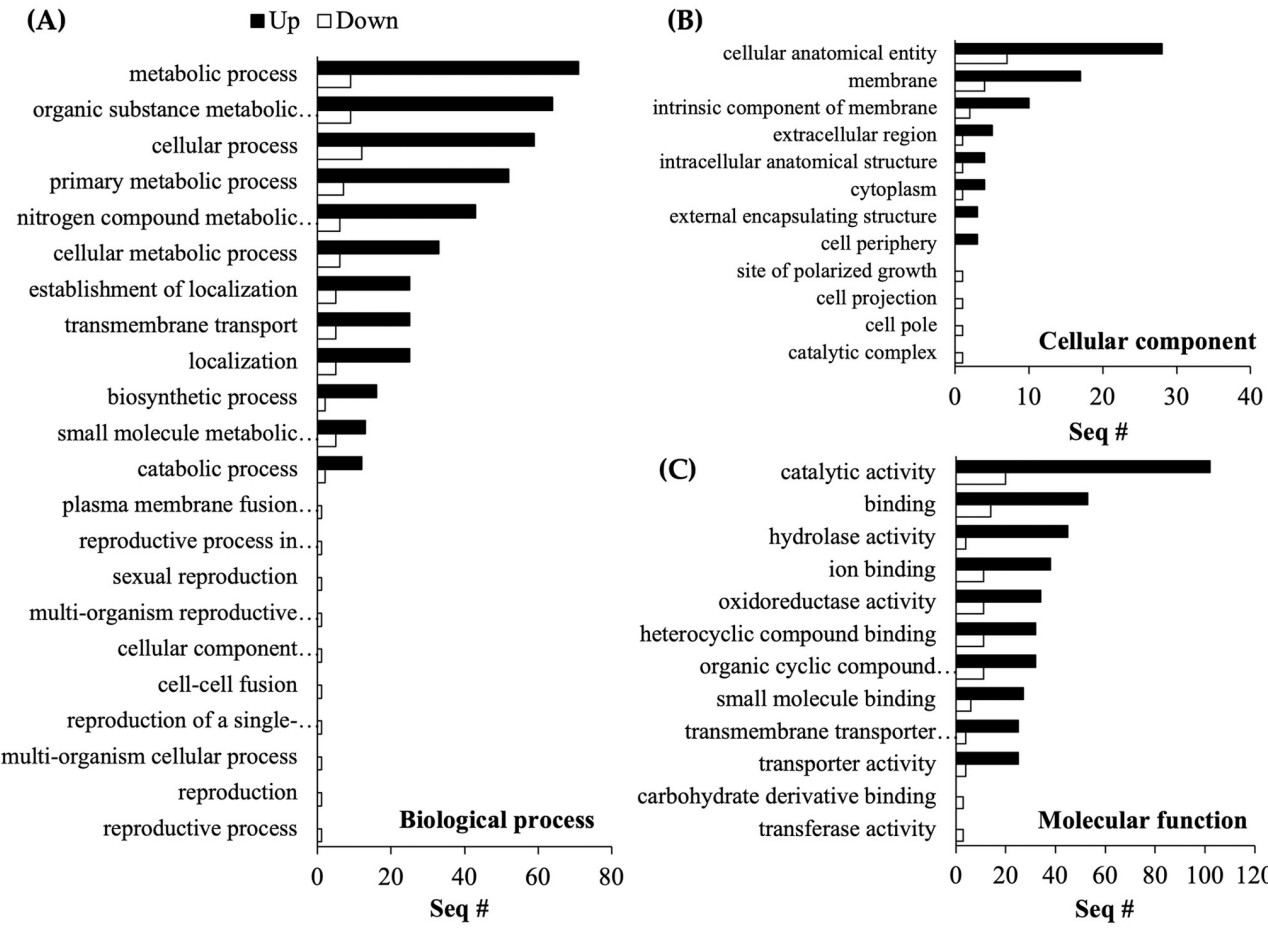

**Fig 2. Gene ontology (GO) analysis of *B. bassiana* JEF-410 DEGs.** Gene Ontology (GO) analysis of non-treated control fungus vs. red mite-infecting fungus was conducted. The up- and down-regulated genes were annotated to a biological process (97 and 15), a cellular component (29 and 7), and molecular function (146 and 27). DEGs of red mite-infecting fungus were annotated in three GO groups: (A) biological process (34.9%), (B) cellular component (11.2%), and (C) molecular function (53.9%).

and organic cyclic compound binding (GO:0097159), and down-regulated genes were annotated to similar GO terms.

## Enriched pathways of *B. bassiana* JEF-410 and red mite

In the infecting *B. bassiana* JEF-410, there were significant changes (FDR < 0.05) in the biosynthesis of secondary metabolites (KEGG:01110), glyoxylate and dicarboxylate metabolism (KEGG:00630), lysin biosynthesis (KEGG:00300), metabolic pathways (KEGG:01100), and tryptophan metabolism pathways (KEGG:00380) by the up-regulated genes and in the metabolic pathways (KEGG:01100) and methane metabolism pathways (KEGG:00680) by the down-regulated genes (Fig 4A and S4 Table). In the infected red mite, there were significant changes (FDR < 0.05) in carbon metabolism (KEGG:01200), citrate cycle (TCA cycle) (KEGG:00020), and propanoate metabolism (KEGG:00640) by the up-regulated genes of red mite (Fig 4B and S5 Table). In validating the gene expression levels of *B. bassiana* JEF-410 and red mites using qRT-PCR, the RNA-seq analysis patterns were confirmed (S1 Fig).

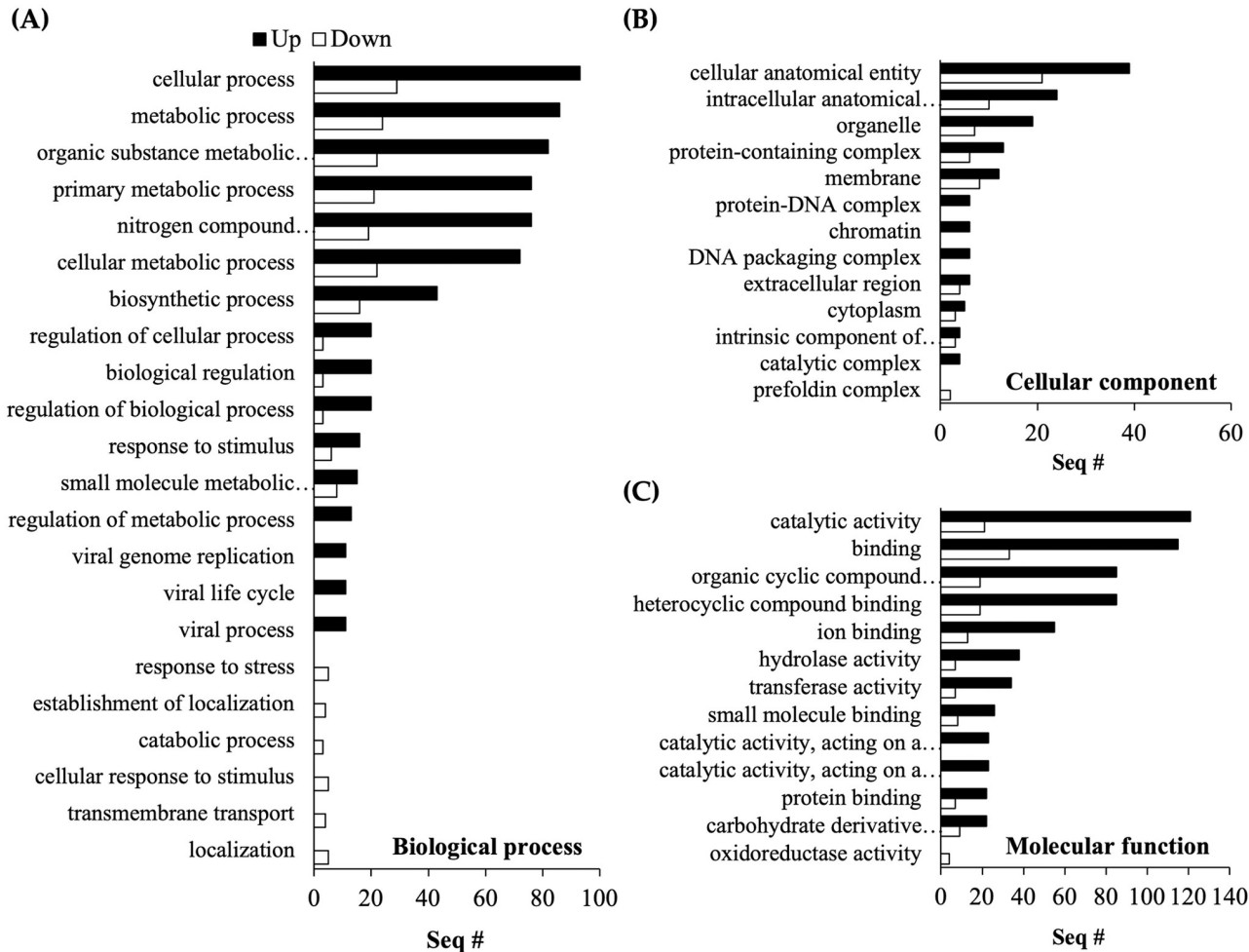

**Fig 3. Gene ontology (GO) analysis of red mite DEGs.** Gene Ontology (GO) analysis of non-treated control red mites vs. fungal-infected red mites was conducted. The up- and down-regulated genes were annotated to a biological process (86 and 3), a cellular component (13 and 21), and molecular function (115 and 21). DEGs of fungal-infected red mites were annotated in three GO groups: (A) biological process (32.7%), (B) cellular component (14.3%), and (C) molecular function (53.0%).

## Secondary metabolite-related genes of *B. bassiana* JEF-410

In the DEGs of *B. bassiana* JEF-410 in the metabolite database of Ascomycota, there were 20 significant genes annotated with e-value lower than 1E-50 (Fig 5). The 20 up-regulated DEGs were classified into four functional groups: enzyme, toxic metabolite biosynthesis, transporter, and energy metabolism. The DEG with enzyme function was lipase A; those involved in toxic metabolite biosynthesis were acyl-CoA ligase easD, acyltransferase easC, amino acid adenylation domain protein, bassianolide nonribosomal peptide synthetase, cytochrome P450 CYP617A1, CYP5293A1, isotrichodermin C-15 hydroxylase, nonribosomal peptide synthetase 7, easA, and O-methylsterigmatocystin oxidoreductase. Transporter DEGs were ABC transporter atnG, ABC transporter C family, MFS transporter, putative HC-toxin efflux carrier TOXA, and riboflavin transporter MCH5; and those if energy metabolism were aromatic amino acid aminotransferase and glucose oxidase.

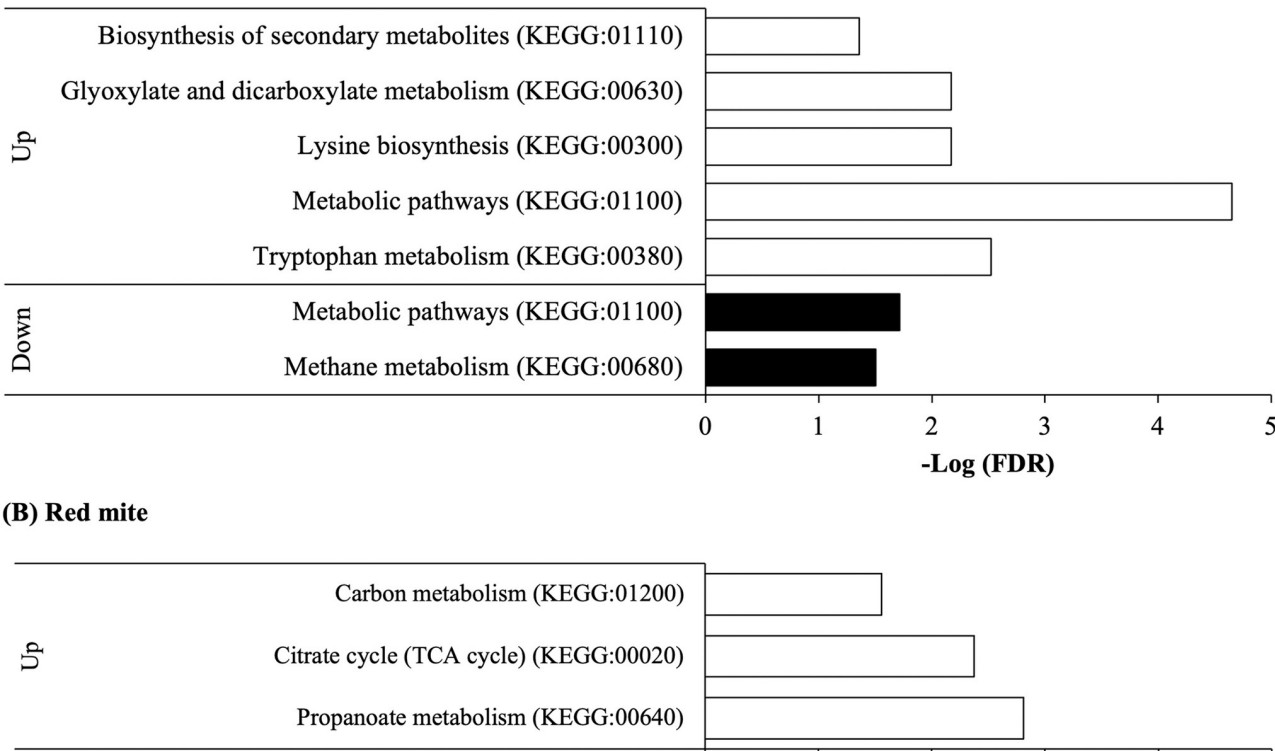

**Fig 4. GO enrichment analysis.** GO enrichment analysis using a Benjamini–Hochberg false discovery rate (FDR) < 0.05 to identify GO terms and KEGG pathways associated with the various groups of genes by g:profiler. (A) Non-treated control fungus vs. red mite-infecting fungus and (B) non-treated control red mites vs. fungal-infected red mites.

### Immune-related genes of red mite

Of the DEGs of red mite in the immune-related gene database of Arthropoda, there were 8 significant genes annotated with e-values lower than 1E-20 (Fig 6). The 10 up-regulated DEGs were classified into three functional groups of physical defense, immune system, and indirect defense. Those involved in physical defense were chitin deacetylase 1, probable nuclear hormone receptor HR38 isoform X7, and spectrin alpha chain; those of the immune system were defensin and E3 ubiquitin-protein ligase CBL-B-B; and those participating in indirect defense were cathepsin L-1, serine proteinase stubble-like isoform X1, serine/threonine-protein kinase SIK3-like isoform X1, negative regulator of reactive oxygen species, elongation factor Tu.

## Discussion

### Fungal GO-enriched pathways at the early stage

In GO enrichment analysis of *B. bassiana* JEF-410, up-regulated genes were enriched in secondary metabolites, glyoxylate and dicarboxylate metabolism, lysin biosynthesis, metabolic pathways, and tryptophan metabolism pathways. The highest values were confirmed in the

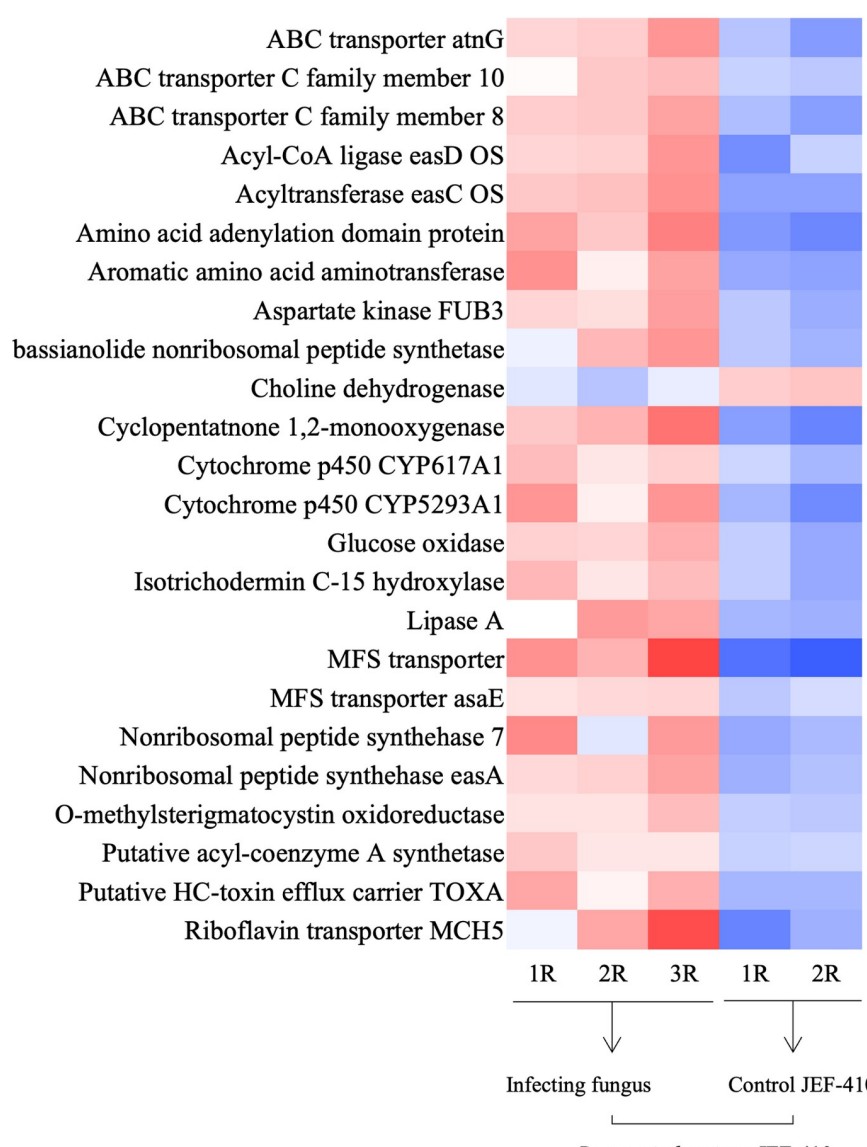

**Fig 5. Heatmap of fungal metabolite-related genes.** Heatmap of secondary metabolite-related genes of *B. bassiana* JEF-410 by significant genes annotated with e-values lower than 1E-50.

metabolic pathway and tryptophan metabolism. Metabolic pathways contain more than 300 types of metabolisms.

In tryptophan metabolism overexpressed in the infecting *B. bassiana* JEF-410, enzymes kynurenine 3-monooxygenase, dihydrolipoyl transsuccinylase, and acetyl-CoA C-

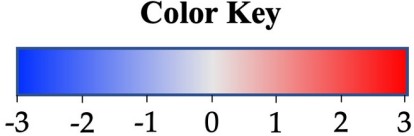

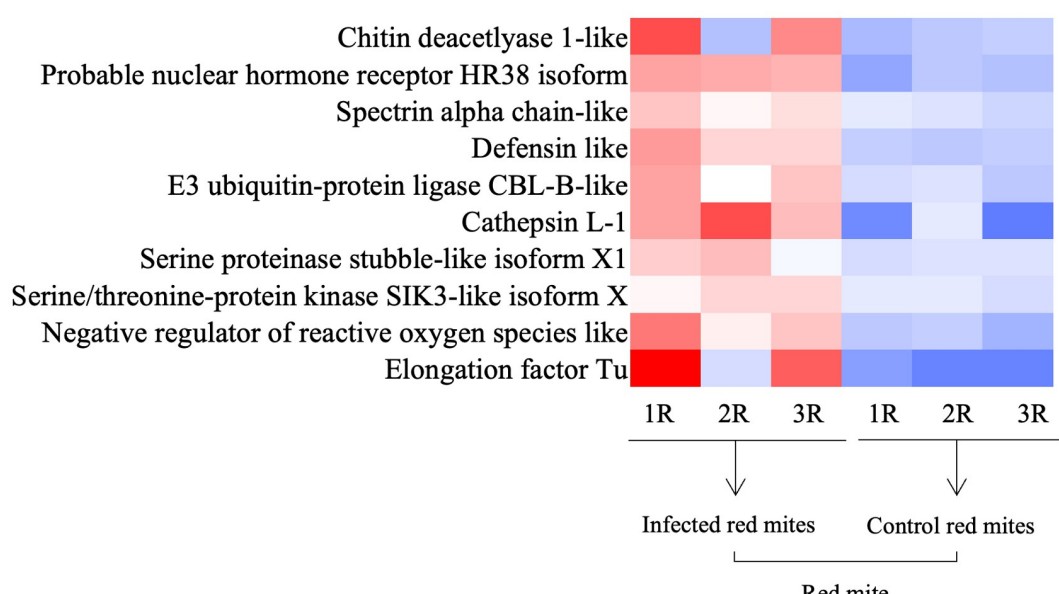

**Fig 6. Heatmap of red mite immune-related genes.** Heatmap of immune-related genes of red mite by significant genes annotated with e-values lower than 1E-20.

acetyltransferase related to acetyl-CoA synthesis were highly expressed Each of the overexpressed enzymes had an indirect effect on pathogenicity. However, acetyl-CoA plays an important role not only in stress regulation, but also in host infection as a precursor or through chromosomal acetylation [25, 26]. The main function of acetyl-CoA is to produce energy after oxidation-reduction through the TCA cycle in mitochondria. In other word, as presented in GO enrichment analysis of Fig 4, the most up-regulated pathways in the infecting JEF-410 were metabolic pathway (KEGG: 01100), followed by tryptophan metabolism (KEGG:00380). Significantly up-regulated genes in the tryptophan metabolism were YBL098W, YIL164C, YGR088W, and YGL202W. Each gene performs the functions of kynurenine 3-monooxygenase, nitrilase, and cytosolic catalase T. The tryptophan metabolism performs energy production by synthesizing kynurenine followed by NAD synthesis or by synthesizing Acetyl-CoA. It was confirmed that the related pathway was up-regulated by validation here in this work. Redox reactions that occur in mitochondria cause reactions such as resistance and lifespan extension [26]. The reason why the redox reaction in mitochondria is important is that, when a fungus infects a host, the host body is anaerobic and stressed. When fungi overcome anaerobic stress, they can easily obtain nutrients from the host [25]. According to previous studies, the HOG1 MAP kinase pathway overcomes oxidative stress by binding nicotinamide adenine dinucleotide (phosphate) NAD(P) to the target site [25]. Additionally, acetyl-CoA induces the chromatin (histone) acetylation reaction in the cell and participates in biochemical reactions [26]. Histone acetylation is an acetylation reaction that occurs in the N-terminal tail of the

histone core of a nucleosome and regulates the expression level of a gene. Histone acetylation is involved in several cellular processes including cell proliferation, cell differentiation, and stress tolerance [26]. Acetyl-CoA synthesis, which is involved in cuticle invasion at the early stage of infection, is related to chromatin/histone acetylation. This synthesis has an important effect on host infection [25]. In other words, when *B. bassiana* JEF-410 infected the red mites, tryptophan metabolism-related pathways were overexpressed. This can overcome host oxidative stress and host physical defense through histone acetylation by synthesizing acetyl-CoA and is involved in various cellular processes required for infection.

The secondary metabolite synthesis process, which showed the most significant change, was also analyzed. The up-regulated genes of *B. bassiana* JEF-410 that invaded red mites were closely related to biosynthesis of secondary metabolites. However, since this pathway identified in the enrichment analysis contains many metabolic pathways, it has been difficult to determine which specific secondary metabolites are biosynthesized in *B. bassiana*. For this reason, metabolite-related genes of *B. bassiana* JEF-410 were found by identifying the DEGs of *B. bassiana* JEF-410 in the metabolite database registered in Uniprot.

## Infection-related genes of *B. bassiana* JEF-410 infecting red mites

The secondary metabolite synthesis process, which confirmed significant changes in up-regulated genes, contained too many metabolic pathways. Therefore, metabolite-related genes were identified using the Uniprot database. The 20 most highly up-regulated genes were identified as metabolite related and were involved in cuticle degradation enzyme, arthropod toxic metabolites, stress response, metabolite secretion related pumps, and energy metabolism [27, 28]. The entomopathogenic fungi secretes various enzymes to invade the host, and in our results, the lipase gene of *B. bassiana* JEF-410, an enzyme for attacking the host cuticle, was up-regulated. Lipase is an enzyme that hydrolyzes long chain triglycerides to fatty acids and acylglycerols and plays an important role in host cuticle degradation [29]. Red mites have an exoskeleton made of polymer chitin, and the epicuticle consists of a layer of wax that limits moisture loss from the outside and a cement layer that protects it from the outside [30, 31]. The increased expression of lipase confirms that *B. bassiana* JEF-410 invades red mites by breaking down the epicuticle.

The fungi that invade the cuticle attack the host attack the host with a variety of substances. In our study, the up-regulated Acyl-CoA ligase easD, acyltransferase easC, non-ribosomal peptide synthetase (NRPS) 7, and NRPS easA genes are clusters that mediate the biosynthesis of beauverolide and emericellamides, which act as antibiotics. The acyl-CoA ligase easD converts the polyketide carboxylic acids released by synthase easB to CoA thioesters. Then, the substrate is loaded into the acyltransferase easC and transferred to the first thiolation (T) domain of the NRPS easA, and easA then condenses the polyketide with one glycine, two alanine, one valine, and one leucine residues, producing emericellamides in the final step of cyclization [32]. The emericellamides are antibiotics produced by *Emericella nidulans* and are similar to the biosynthetic pathway of cyclodepsipeptide beauveriolides in *Beauveria* sp. [32]. In addition, the amino acid adenylation domain protein plays a role in the specific recognition and activation of amino acids into adenylyl amino acids, followed by the previously described T-domain [33]. This suggests that *B. bassiana* JEF-410 in red mites produces beauveriolides. In addition, the up-regulated P450s genes, CYP167A1 and CYP5293A1, are involved in biosynthesis of beauvericin and bassianolide. The CYP5293 family is involved in beauvericin biosynthesis [34], and the CYP617 family is involved in non-ribosomal peptide synthetase for bassianolide biosynthesis [35] and fatty acid degradation associated with insect hydrocarbon degradation [36, 37]. Up-regulation of the bassianolide nonribosomal peptide synthetase gene supports the

biosynthesis of bassianolide by *B. bassiana* JEF-410 in red mites. This suggests that P450s, which play various roles in fungi, produce host-selective toxicity and entomopathogenic metabolites such as beauvericin and bassianolide in *B. bassiana* JEF-410 that invade red mites [36–38].

It is likely that the increase in gene expression of secondary metabolites such as beauvericin, beauveriolides, and bassianolide of *B. bassiana* JEF-410 is probably involved in the infection of red mites. Beauvericin was first isolated from *B. bassiana* and exhibits various biological activities such as antibacterial and antitumor activity [39, 40] and contributes to pathogenicity by acting as a virulence factor regulating the Arthropoda immune system [41]. The metabolite has acaricidal activity not only toward insects but also toward mites such as *Tetranychus urticae* and *Sarcoptes scabiei* [42, 43]. In this study, expression of the P450 CYP5293 family for beauvericin biosynthesis in *B. bassiana* JEF-410 in red mites was up-regulated, which is predicted to increase the biosynthesis of beauvericin. In other words, it is indicated that *B. bassiana* JEF-410 provides high toxicity to red mites through production of beauvericin, a major acaricidal metabolite. Beauveriolide was first identified in *Beauveria tenella*; currently, 28 beauveriolides that have some degree of insecticidal activity have been reported in other *Beauveria* sp. and *Cordyceps* sp. [44–46].

In our results, the genes involved in the biosynthesis of beauveriolide were overall up-regulated, and it is expected that the secretion of beauveriolide is stimulated during the mite invasion process. Beauveriolide has been reported to have some insecticidal activity in other arthropods and is likely to have a similar function in red mites. However, due to the lack of reports on the acaricidal activity and function of beauveriolide against red mites, a functional study of this metabolite is needed. Also, bassianolide acts as a highly virulent factor of *B. bassiana* against the host [34]. In our study, the increase in secretion is posited to be due to up-regulation of the CYP617 family and bassianolide nonribosomal peptide synthetase genes. As with beauveriolide, there is a lack of study on virulence and function of bassianolide against mites, and there is a need to study whether secretion is stimulated during infection of red mites by *B. bassiana* JEF-410. The results of our study suggest beauveriolide and bassianolide as candidate acaricidal metabolites of *B. bassiana* against red mites, which were not well known before. In addition, O-methylsterigmatocystin oxidoreductase involved in aflatoxin biosynthesis [47] and isotrichodermin C-15 hydroxylase involved in the trichothecenes biosynthetic pathway [48] were up-regulated, suggesting that aflatoxin and trichothecene act as toxic substances to red mites. In particular, the trichothecenes are sesquiterpene toxins produced by various fungal species, primarily Sordariomycetes fungi [49]. The trichothecene from *Isaria tenuipes* has been reported to be toxic to insects [50]. Although genome analysis in the genera *Beauveria* and *Cordyceps* of Sordariomycetes revealed the presence of homologues in the trichothecenes biosynthetic gene cluster [51], they have not been reported to produce these substances [52]. However, our results suggest that red mite-invading *B. bassiana* JEF-410 produces trichothecenes.

Along with the production of toxic metabolites, the activity of transporters to secrete them also appears to increase. In *B. bassiana* JEF-410 in red mites, the major facilitator superfamily (MFS) and ATP-binding cassette transporter (ATP) transporter genes for transporting the toxin metabolites were also up-regulated. *B. bassiana* contains 13 unique genes involved in transport, comprising five that belong to MFS families, four of ABC families, dedicated cytophore transporter, dicarboxylate transporter, sodium antiporter, and proton antiporter [53]. The up-regulated MFS transporter genes of *B. bassiana* JEF-410 in red mites were involved in substrate and nutrient sensing, amino acid and peptide receptors, and transport of ATP-binding cassette transporters [54]. In addition, the MFS transporter of *B. bassiana* is a part of the gene mediating the biosynthesis of substances necessary for toxicity in the host body and

known to biosynthesize oosporein [55]. This transporter also acts as an efflux pump required for efficient secretion of produced fungal secondary metabolites [56]. It can be suggested that the up-regulated MFS transporter genes of *B. bassiana* JEF-410 promote the production of metabolites that are toxic to the host and the activity of the transporting toxic metabolites described above. The ABC transporter is also involved in the transport of metabolites; generally, the ABC transporter uses ATP to transport various substrates such as lipids, pheromones, heavy metals, and xenobiotics across biological membranes [57]. In addition, this transporter is involved in multidrug resistance to fungicides and secretion of antibiotics and toxins [53].

In *B. bassiana*, the ABC transporter was found to be associated with fungicide resistance, oxidative stress resistance, and toxicity [58]. The up-regulated ABC transporter family genes of *B. bassiana* JEF-410 in red mites have been reported to have an association with multidrug resistance [59]. Thus, it is predicted that, in red mites, *B. bassiana* JEF-410 utilizes ABC transporters to resist external stress as well as substrate transport. In addition, the putative HC-toxin efflux carrier TOXA that acts in self-protection against toxins or as an efflux pump that releases toxins into the extracellular environment [60] and the riboflavin transporter associated with riboflavin uptake [61] were up-regulated. This suggests that their respective functions also occur in *B. bassiana* JEF-410 when invading red mites.

The generation of energy needed by *B. bassiana* JEF-410 to invade the red mites also increased. Aromatic amino acid aminotransferase is mainly involved in the degradation of tryptophan. It catalyzes the formation of methionine from 2-keto-4-methylthiobutyrate (KMTB) in the methionine salvage pathway using aromatic amino acids such as tyrosine, phenylalanine, and tryptophan as amino donors. It catalyzes the irreversible transamination of the L-tryptophan metabolite L-kynurenine to form kynurenic acid (KA) using pyruvate acid as its amino acceptor [62–65]. In our enrichment analysis results, the tryptophan metabolism pathway of *B. bassiana* JEF-410 was affected by up-regulated DEGs, and it is predicted that the pathway is involved by degradation of tryptophan by the aminotransferase. Glucose oxidase (GOD) is a flavoprotein involved in catalysis of the oxidation of glucose to lactones and h2 using molecular oxygen as an electron acceptor [66, 67]. It is thought to be an oxidase enzyme for *B. bassiana* JEF-410 to break down glucose in the red mite and use it for nutrients. Also, it is speculated that energy metabolism is actively taking place due to degradation of tryptophan.

## Immune-related genes of red mites invaded by *B. bassiana* JEF-410

The up-regulated genes of the red mites infected by *B. bassiana* JEF-410 were closely related to immune-related gene group. However, this pathway identified in the enrichment analysis is involved in carbon metabolism, citrate cycle (TCA cycle), and propanoate metabolism. For this reason, immune-related genes of Arthropoda were found by identifying the DEGs of red mite in the metabolite database registered in Uniprot. The 20 most highly up-regulated genes were identified as immune related and were involved in physical changes in cuticle and secretion of defense substances. The nine up-regulated DEGs of *B. bassiana* JEF-410-infected red mites were annotated to innate immune and digestion-related genes in the immune-related gene database of Arthropoda.

It is predicted that red mites primarily defend against fungal invasion by physical changes in the cuticle. The up-regulated chitin deacetylase 1 is responsible for chitin deacetylation [68] and plays an important role in cuticle strengthening and the interaction between chitin and binding proteins through the conversion of chitin to chitosan [69]. It is suggested that red mites defend against *B. bassiana* by strengthening the cuticle layer. In addition, they also might evade the fungus through molting. Nuclear hormone receptor HR78 binds directly to

the repeats of the 5'-AGGTCA-3' sequence to inhibit the ecdysone response and is essential for regulating molting during larval development [70, 71]. It is expected that the molting pattern of the red mites is affected to some extent by *B. bassiana*. The red mites not only strengthened the cuticle, but also increased the expression of genes for maintaining cell tissue. The spectrin alpha chain gene is essential for the survival and development of larvae in Drosophila and plays a role in stabilizing cell-to-cell interactions that are important for intracellular shape and intracellular tissue [72]. In other words, it could be suggested that the red mites attacked by *B. bassiana* defend themselves by strengthening and maintaining cell tissues.

When *B. bassiana* penetrates the fortified cuticle, immune systems act inside the red mites. The main defense factor of red mites is defensin, which is structurally similar to insect defensin and other microplusins. Defensin is a major immune response in Arthropoda and plays an important role in defense against bacteria and fungi [73–75]. Defensin was reported to be expressed mainly in fat body and midgut epithelial cells in insects [76, 77] and in hemocytes and midgut in mites [78–81]. This suggests a similar response between mites and insects, and our results are examples of a major immune response to the entomopathogenic fungus *B. bassiana*. In addition, E3 ubiquitin-protein ligase is an important factor in regulatory mechanisms in innate and adaptive immunity in mammals [82, 83]. Although its activity in mites is not clear, it is suggested that it may be necessary for immune function.

Unlike direct defense, up-regulated genes that are expected to have an indirect defense effect were also confirmed. Digestion-related genes were up-regulated inside the red mites. In particular, cathepsin [84, 85] and serine protease [86] were involved in hemoglobin and nutrient digestion. Although those are not directly related to fungi, it is likely that mites provide the necessary elements for immunity to fungi through digestion. The negative regulator of reactive oxygen species (NRROS) limits reactive oxygen species (ROS) generation and may regulate NOX2 expression and ROS generation by regulating the expression of NRROS [87, 88]. Reduction of ROS may be detrimental to fungal defense, but it could induce an increase in other symbiotic microorganisms. This may be an indirect defense using symbiotic microorganisms. In addition, elongation factor Tu (EF-Tu), which transports aminoacylated tRNA to the ribosome [89], was up-regulated.

## Conclusions

The results of our study describe the overall process that occurs in the interaction between *B. bassiana* and red mites. The lack of information about the interaction of the two organisms can be supplemented with these results. The red mite-infecting *B. bassiana* JEF-410 up-regulated the expression of genes in energy metabolism and secondary metabolite biosynthesis pathways. In particular, in tryptophan metabolism, genes synthesizing acetyl-CoA from tryptophan were up-regulated. The synthesized acetyl-CoA becomes a back-bond for energy metabolism, histone acetylation, and various secondary metabolites. Tryptophan metabolism was the dominant factor in energy metabolism and was associated with host-microbial interactions. In addition, *B. bassiana* JEF-410 produced various secondary metabolites to invade red mites. It primarily decomposes and penetrates the cuticle of red mites using enzymes such as lipase, and the invading strain is presumed to have acaricidal activity against red mites by secreting metabolites toxic to arthropods such as beauvericin, bassianolide, and beauveriolide. Arthropod toxins are synthesized as polyketides synthesized with acetyl-CoA as a back-bond. This suggests that up-regulation of tryptophan metabolism is involved in the synthesis of various secondary metabolites (Fig 7).

To produce these metabolites, it is likely that the activation of protein production and metabolic pathways to obtain energy increases the acaricidal activity of *B. bassiana* JEF-410. On the other hand, the results suggest that *B. bassiana* JEF-410-infected red mites are physically

**Fig 7. Analysis of enriched of tryptophan metabolism pathway that *B. bassiana* JEF-410 were infected red mite were infected at early stage.** (A) Summarized acetyl-CoA synthesis pathway and secondary metabolite synthesis pathway of BNA2 dioxygenase, BNA4 kynurenine 3-monooxygenase, BNA7 arylformamidase, PEP5 tethering complex subunit, KGD2 alpha-ketoglutarate dehydrogenase, LPB3 acetyl-CoA C-acetyltransferase; PK, polyketide. (B) Histone acetylation is responsible for synthesis of acetyl-CoA and amplification of DNA expression level. (C) The Hog1 pathway for overcoming oxidative stress in mitochondria starts from acetyl-CoA; MAPK mitogen-activated protein kinase; Hog1 branched MAPK signal transduction system.

defended by strengthening the cuticle and maintaining the cell tissue, and that the defensin-led immune response was performed. Metabolic activity to produce energy necessary for defense has been confirmed in red mites, and it is speculated that energy metabolism plays an important role in the interaction between host and microorganisms in addition to direct toxic

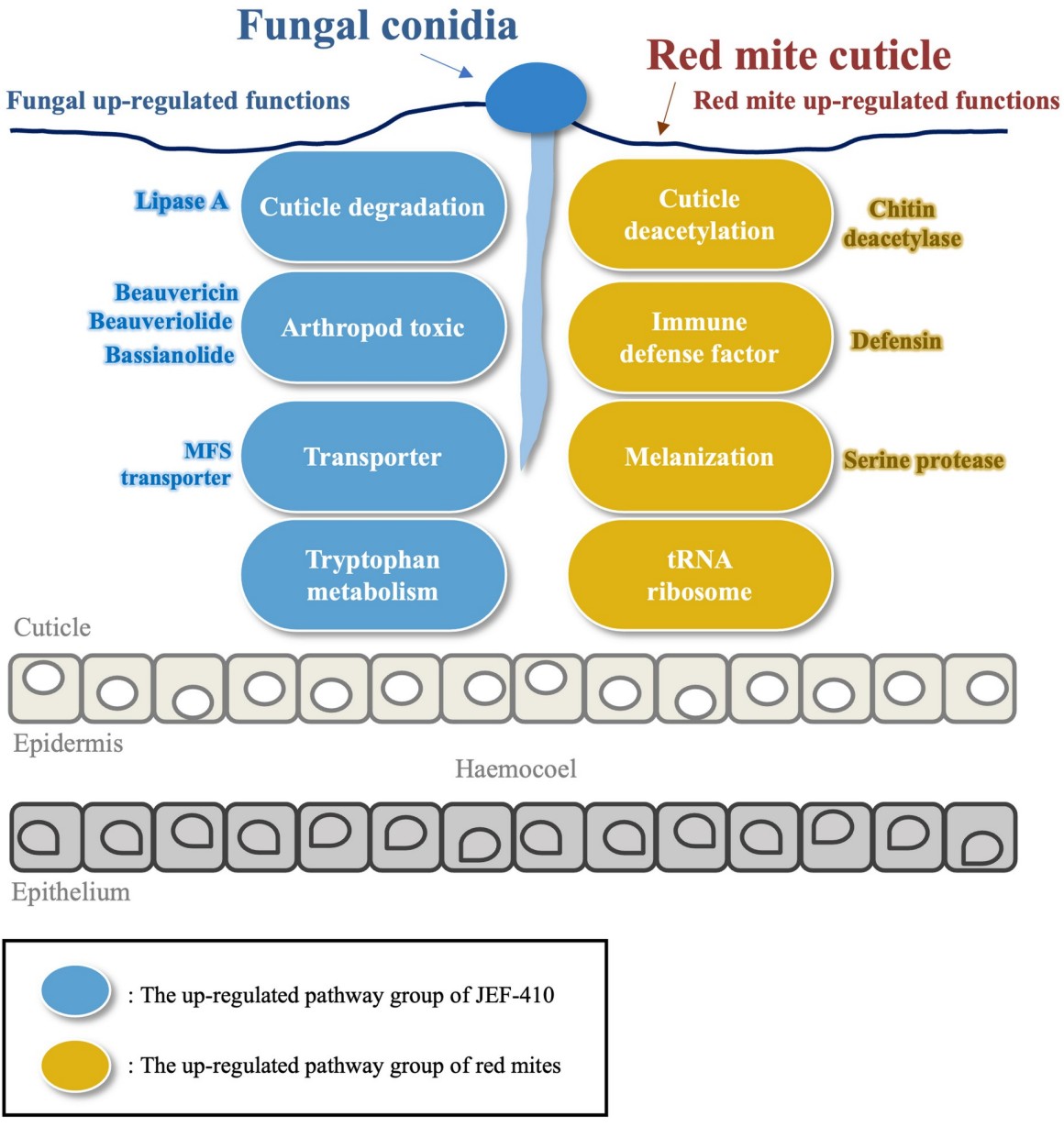

**Fig 8. Interaction of *B. bassiana* JEF-410 and red mite at early stage of infection.**

metabolites or defense substances (Fig 8). Finally, an essential element in the interaction between *B. bassiana* and red mites was proposed, which will improve the value of fungi with important data for elucidating the mode of action of entomopathogenic fungi as biological pesticides in the future.

## Supporting information

**S1 Fig. Validation for RNA-seq analysis by qRT-PCR.**
(PPTX)

**S1 Table. Primers used in qRT-PCR for validation of RNA-sequencing.**
(PPTX)

**S2 Table. Summary of Illumina sequences.**
(PPTX)

**S3 Table. *De novo* assembly of red mite sequences.**
(PPTX)

**S4 Table. KEGG pathway of infecting *B. bassiana* JEF-410 DEGs.**
(PPTX)

**S5 Table. KEGG pathway of infected red mite DEGs.**
(PPTX)

## Acknowledgments

This work was performed at Jeonbuk National University insect microbiology and biotechnology laboratory (IMBL). The group of insect microbiology and biotechnology laboratory was as follows: Jae Su Kim (principal Investigator, E-mail: jskim10@jbnu.ac.kr, Jeonbuk national university, Jeonju; So Eun Park, Jeonbuk national university, Jeonju; Jong-Cheol Kim, Jeonbuk national university, Jeonju; Yeram Im, Jeonbuk national university, Jeonju; Insoo Jeon, Jeonbuk national university, Jeonju; Yujin Jung, Jeonbuk national university, Jeonju; Yulim Park, Jeonbuk national university, Jeonju; Kijung Kim, Jeonbuk national university, Jeonju; Gahyeon Song, Jeonbuk national university, Jeonju; Hanna Yang, Jeonbuk national university, Jeonju; Novadhea Salsabilla, Jeonbuk national university, Jeonju.

## Author Contributions

**Conceptualization:** Jae Su Kim.

**Formal analysis:** So Eun Park, Jong-Cheol Kim.

**Investigation:** So Eun Park, Jong-Cheol Kim, Yeram Im, Jae Su Kim.

**Resources:** So Eun Park, Yeram Im.

**Software:** So Eun Park.

**Supervision:** Jae Su Kim.

**Validation:** So Eun Park, Jong-Cheol Kim.

**Visualization:** So Eun Park.

**Writing – original draft:** So Eun Park, Jae Su Kim.

**Writing – review & editing:** So Eun Park, Jong-Cheol Kim, Jae Su Kim.

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
