## [Decision Letter · Decision Letter 0]

6 Nov 2022

PONE-D-22-27296Pathogenesis and defense mechanism while Beauveria bassiana JEF-410 infects poultry red mite, Dermanyssus gallinaePLOS ONE

Dear Dr. Kim,

Thank you for submitting your manuscript to PLOS ONE. After careful consideration, we feel that it has merit but does not fully meet PLOS ONE’s publication criteria as it currently stands. Therefore, we invite you to submit a revised version of the manuscript that addresses the points raised during the review process.

The manuscript is discuss an interesting point but contains some issues as addressed by the reviewers.  

We look forward to receiving your revised manuscript.

Kind regards,

Shawky M Aboelhadid, PhD

Academic Editor

PLOS ONE

Journal Requirements:

"This work was supported by the National Research Foundation of Korea(NRF) grant funded by the Korea government(MSIT) (NRF-2018R1A2B6001351)."

Please state what role the funders took in the study.  If the funders had no role, please state: ""The funders had no role in study design, data collection and analysis, decision to publish, or preparation of the manuscript."" If this statement is not correct you must amend it as needed. 

"This work was supported by the National Research Foundation of Korea(NRF) grant funded by the Korea government(MSIT) (NRF-2018R1A2B6001351)"

"This work was supported by the National Research Foundation of Korea(NRF) grant funded by the Korea government(MSIT) (NRF-2018R1A2B6001351)."

Reviewers' comments:

Reviewer's Responses to Questions

**Comments to the Author**

1. Is the manuscript technically sound, and do the data support the conclusions?

Reviewer #1: Partly

Reviewer #2: Yes

2. Has the statistical analysis been performed appropriately and rigorously? 

Reviewer #1: Yes

Reviewer #2: Yes

3. Have the authors made all data underlying the findings in their manuscript fully available?

Reviewer #1: Yes

Reviewer #2: Yes

4. Is the manuscript presented in an intelligible fashion and written in standard English?

Reviewer #1: Yes

Reviewer #2: Yes

5. Review Comments to the Author

Reviewer #1: The work is novel and explores the interactions that exist between the entomopathogenic fungus and its host red mites. By elucidating these interactions, it is possible to optimize the insecticidal action of the fungus, either by inducing or overexpressing insecticidal agents or by reducing the insect's defense tools.

However, I have some observations:

1. The introduction adequately contextualizes the red mite pest. However, it does not clearly indicate which are the advances described in the literature about the interactions between entomopathogenic fungus and insect.

1.1. In lines 68-70, a work is cited that studies the transcriptome of two-spotted spider mite, however, this study is about the expression of genes in diapause. Therefore, it does not seem pertinent to cite it without informing that it is in this specific state. Besides, red mites do not go through this stage.

1.2 The pathogenicity mechanisms of the fungus and the defense mechanisms of the insect should be specified in order to provide the reader with guidelines for the work. What factors does the fungus secrete? What are the defense mechanisms of the insect?

2. The methodology is very well explained and the statistical analyses correspond to the type of sample.

However:

2.1. In lines: 97, 98, 120, 127, 203, 203, 204, 205, 209, 210 and 211, separate degrees from the unit celsius (e.g. 20° C).

2.2. On line 166 correct URL: https://bioconductor.org/packages/devel/bioc/html/edgeR.html.

3. The results are well written, just consider these remarks:

3.1 In line 274: add period after 53.0%.

3.2 In Figure 1C, 5 and 6: add meaning of the abbreviation "1R", "2R", "3R".

3.3 In Figure 3: complete the sentences, because it is necessary to specify what each of the evaluated gene groups refers to. The figure should provide all the information by itself.

4. With respect to the results, I consider that some of the conclusions are somewhat hasty. In addition, the information that is taken for granted was not referenced.

4.1. Line 330: The title does not cover what is addressed in the body of the text. If you intend to establish that the mechanism of tryptophan is in the early stages of infection.... What is the range or stages that this encompasses? It should be contextualized as to what "early stage" refers to. Personally, I would not state it as part of the title of the result, since one could not be sure that it is in the early stage. That is mostly a conclusion formulated from the literature, but not from the experiment itself.

4.2. In line 335-335, I would eliminate this sentence, since it is arbitrary. Neither in the introduction or results have they pointed out why tryptophan metabolism is relevant. I consider that eliminating it and starting with the following paragraph is the most appropriate.

4.3. In lines 340-347, add a reference that evidences what has been pointed out.

4.4 In line 380: I would not use "Suggests", since the mechanism of infection is highly evidenced, especially of Beauveria bassiana. So it is a fact that it invades the hemolymph, by penetration of the epicuticle. Use the word "confirms" instead.

4.5 On line 382: Change sentence to "The fungi that invade the cuticle attack the host...".

4.6 On line 452: The conclusion seems unfounded, since one could not speak of promoting. Only that in conjunction with the increase of genes coding for secondary metabolites, it could be established that secondary metabolites are relevant for the infection of the fungus in red mites. The word promote implies temporal order, and neither result accounts for that.

Reviewer #2: In this paper，the pathogenesis of the acaricidal fungus Beauveria bassiana JEF-410 and the

defense mechanisms of red mites were studied by transcriptome analysis. Although the results presented here are intriguing, numerous grave issues still exist.

Major concerns:

1. The strain name needs to be consistently shortened throughout the text. JEF-410 is referred to in lines 25, 28, and 39, but B. bassiana JEF-410 is found in line 29.

2. Lines 68-70, reference 14 should be added at the end of the sentence “suppression of up-regulated genes suggests enhanced insecticidal activity.” However, I couldn’t find the relative information as this paper described in reference 14, please check again.

3. The bar chart's color in figures 2, 3, and 4 is not appealing.

4. Lines 240-249, results regarding the heat map of B. bassiana JEF-410 and red mite (Fig 1C) were inconclusive. Additionally, certain information should be included in the figure1 illustration, such as descriptions of 1R, 2R, and 3R. Why is the B. bassiana JEF-410 control group experiment lacking in 3R data?

5. Line 269, “In the GO analysis of red mite,” should be red mites.

6. Line 340, the author claimed that overexpressed enzymes had no significant effect on pathogenicity, although analysis revealed that synthesizing acetyl-CoA and is involved in various cellular processes required for infection (lines 358-359). Some of the conclusions are in disagreement. Please elaborate.

7. Relevant references should be added to lines 340–344.

8. More detailed information has to be provided because the subject on this page is somewhat broad. Instead of simply discussing their metabolic pathways in general, what are acetyl-CoA and certain energy metabolism-related genes in B. bassiana JEF-410 infecting red mites, and how frequently are they up-regulated or down-regulated?

6. PLOS authors have the option to publish the peer review history of their article (what does this mean?). If published, this will include your full peer review and any attached files.

Reviewer #1: **Yes: **Matias Arias-Aravena

Reviewer #2: No

---

## [Author Response · Author response to Decision Letter 0]

18 Dec 2022

Reviewer's Responses to Questions

Review Comments to the Author

→ I really appreciate your comments and recommendation. Per your comments I revised this manuscript very carefully.

Reviewer #1: The work is novel and explores the interactions that exist between the entomopathogenic fungus and its host red mites. By elucidating these interactions, it is possible to optimize the insecticidal action of the fungus, either by inducing or overexpressing insecticidal agents or by reducing the insect's defense tools.

However, I have some observations:

→ Thank you! As commented, I described these interactions much clear in the revised manuscript. 

1. The introduction adequately contextualizes the red mite pest. However, it does not clearly indicate which are the advances described in the literature about the interactions between entomopathogenic fungus and insect.

1.1. In lines 68-70, a work is cited that studies the transcriptome of two-spotted spider mite, however, this study is about the expression of genes in diapause. Therefore, it does not seem pertinent to cite it without informing that it is in this specific state. Besides, red mites do not go through this stage.

→ As commented, the description of the two-spotted spider mite were deleted and changed with more suitable information.

1.2 The pathogenicity mechanisms of the fungus and the defense mechanisms of the insect should be specified in order to provide the reader with guidelines for the work. What factors does the fungus secrete? What are the defense mechanisms of the insect?

→ As commented, the description was modified and updated with a resistance mechanism of tick against fungal pathogenesis during attack.

2. The methodology is very well explained and the statistical analyses correspond to the type of sample.

However:

2.1. In lines: 97, 98, 120, 127, 203, 203, 204, 205, 209, 210 and 211, separate degrees from the unit celsius (e.g. 20° C).

→ As commented, I changed all symbols to °C.

2.2. On line 166 correct URL: https://bioconductor.org/packages/devel/bioc/html/edgeR.html.

→ As commented, I changed the URL address. 

3. The results are well written, just consider these remarks:

3.1 In line 274: add period after 53.0%. 

→ As commented, I corrected it with period marking. 

3.2 In Figure 1C, 5 and 6: add meaning of the abbreviation "1R", "2R", "3R".

→ As commented, I added the meaning of “R” on figure 1: The meaning of the ‘R’ attached to each number is the repetition of the sample.

3.3 In Figure 3: complete the sentences, because it is necessary to specify what each of the evaluated gene groups refers to. The figure should provide all the information by itself.

→ As commented, I updated the figure legends in figure 3 and 4 to provide the complete information in the sentence itself.

4. With respect to the results, I consider that some of the conclusions are somewhat hasty. In addition, the information that is taken for granted was not referenced.

4.1. Line 330: The title does not cover what is addressed in the body of the text. If you intend to establish that the mechanism of tryptophan is in the early stages of infection.... What is the range or stages that this encompasses? It should be contextualized as to what "early stage" refers to. Personally, I would not state it as part of the title of the result, since one could not be sure that it is in the early stage. That is mostly a conclusion formulated from the literature, but not from the experiment itself.

→ In our previous experiment (published data at https://doi.org/10.1007/s10526-021-10110-w (Park et al., BioControl, 2021)), LT25 of B. bassiana JEF 410 at 1x107 conidia/ml was about 2.5 days after treatment. Based on this data, we determined it as an early stage of infection. 

4.2. In line 335-335, I would eliminate this sentence, since it is arbitrary. Neither in the introduction or results have they pointed out why tryptophan metabolism is relevant. I consider that eliminating it and starting with the following paragraph is the most appropriate. 

→ As commented, I deleted it and started with the suggestion.

4.3. In lines 340-347, add a reference that evidences what has been pointed out.

→ As commented, I added a reference to support it.

4.4 In line 380: I would not use "Suggests", since the mechanism of infection is highly evidenced, especially of Beauveria bassiana. So it is a fact that it invades the hemolymph, by penetration of the epicuticle. Use the word "confirms" instead.

→ As commented, I revised it.

4.5 On line 382: Change sentence to "The fungi that invade the cuticle attack the host...".

→ As commented, I revised the sentence.

4.6 On line 452: The conclusion seems unfounded, since one could not speak of promoting. Only that in conjunction with the increase of genes coding for secondary metabolites, it could be established that secondary metabolites are relevant for the infection of the fungus in red mites. The word promote implies temporal order, and neither result accounts for that.

→ As you commented, we revised it as follows: It is likely that the increase in gene expression of secondary metabolites such as beauvericin, beauveriolides, and bassianolide of B. bassiana JEF-410 is probably involved in the infection of red mites. 

Reviewer #2: In this paper，the pathogenesis of the acaricidal fungus Beauveria bassiana JEF-410 and the

defense mechanisms of red mites were studied by transcriptome analysis. Although the results presented here are intriguing, numerous grave issues still exist.

Major concerns:

1. The strain name needs to be consistently shortened throughout the text. JEF-410 is referred to in lines 25, 28, and 39, but B. bassiana JEF-410 is found in line 29.

→ As commented, I made the strain name shortened uniformly throughout the revised manuscript.

2. Lines 68-70, reference 14 should be added at the end of the sentence “suppression of up-regulated genes suggests enhanced insecticidal activity.” However, I couldn’t find the relative information as this paper described in reference 14, please check again.

→ As commented, I updated the reference to make it strongly support our data and located at the end of the sentence.

3. The bar chart's color in figures 2, 3, and 4 is not appealing.

→ As commented, I made the figures in black and white mode.

4. Lines 240-249, results regarding the heat map of B. bassiana JEF-410 and red mite (Fig 1C) were inconclusive. Additionally, certain information should be included in the figure1 illustration, such as descriptions of 1R, 2R, and 3R. Why is the B. bassiana JEF-410 control group experiment lacking in 3R data?

→ Your comment could be right, but in this analysis of heatmap I tried to investigate the variation of gene expression among the repetitions in each treatment. And the heatmaps showed a quite similar patterns of expression in each treatment.

→ I sequenced the 3rd replicate (3R) of the non-infection JEF-410 but the gene expression pattern was significantly different from the other two replicates, so unfortunately 3R was exclusive from this analysis. 

5. Line 269, “In the GO analysis of red mite,” should be red mites.

→ As commented, I corrected it as red mites.

6. Line 340, the author claimed that overexpressed enzymes had no significant effect on pathogenicity, although analysis revealed that synthesizing acetyl-CoA and is involved in various cellular processes required for infection (lines 358-359). Some of the conclusions are in disagreement. Please elaborate.

→ Your comment is right. To clarify it, I deleted the sentence (overexpressed enzymes had no significant effect on pathogenicity). 

7. Relevant references should be added to lines 340–344.

→ As commented, I added relevant references.

8. More detailed information has to be provided because the subject on this page is somewhat broad. Instead of simply discussing their metabolic pathways in general, what are acetyl-CoA and certain energy metabolism-related genes in B. bassiana JEF-410 infecting red mites, and how frequently are they up-regulated or down-regulated?

→ Thank you for your good comments! I provided major genes from metabolism for more deep understanding. As presented in GO enrichment analysis of Figure 4, the most up-regulated pathways in the infecting JEF-410 were metabolic pathway (KEGG: 01100), followed by tryptophan metabolism (KEGG:00380). Significantly up-regulated genes in the tryptophan metabolism were YBL098W, YIL164C, YGR088W, and YGL202W. Each gene performs the functions of kynurenine 3-monooxygenase, nitrilase, and cytosolic catalase T. The tryptophan metabolism performs energy production by synthesizing kynurenine followed by NAD synthesis or by synthesizing Acetyl-CoA. It was confirmed that the related pathway was up-regulated by validation here in this work. 

→ In this work, I extracted a total of RNA in 2.5 days after fungal treatment, so I could just describe the up- and down-regulation in the determined time point. However, as the pathogenesis is proceeding, the pathogenesis-related genes would be more highly up-regulated and finally back to normal expression

---

## [Decision Letter · Decision Letter 1]

28 Dec 2022

Pathogenesis and defense mechanism while Beauveria bassiana JEF-410 infects poultry red mite, Dermanyssus gallinae

PONE-D-22-27296R1

Dear Dr. Jea Su Kim, 

We’re pleased to inform you that your manuscript has been judged scientifically suitable for publication and will be formally accepted for publication once it meets all outstanding technical requirements.

Kind regards,

Shawky M Aboelhadid, PhD

Academic Editor

PLOS ONE

Additional Editor Comments (optional):

Reviewers' comments:

Reviewer's Responses to Questions

**Comments to the Author**

1. If the authors have adequately addressed your comments raised in a previous round of review and you feel that this manuscript is now acceptable for publication, you may indicate that here to bypass the “Comments to the Author” section, enter your conflict of interest statement in the “Confidential to Editor” section, and submit your "Accept" recommendation.

Reviewer #1: All comments have been addressed

Reviewer #2: All comments have been addressed

2. Is the manuscript technically sound, and do the data support the conclusions?

Reviewer #1: Yes

Reviewer #2: Yes

3. Has the statistical analysis been performed appropriately and rigorously? 

Reviewer #1: Yes

Reviewer #2: Yes

4. Have the authors made all data underlying the findings in their manuscript fully available?

Reviewer #1: Yes

Reviewer #2: Yes

5. Is the manuscript presented in an intelligible fashion and written in standard English?

Reviewer #1: Yes

Reviewer #2: Yes

6. Review Comments to the Author

Reviewer #1: Dear,

the writing, with the corrections, tells the story much better. Good job!.

The introduction provides relevant background and the conclusions are correct with respect to the work done.

Reviewer #2: (No Response)

7. PLOS authors have the option to publish the peer review history of their article (what does this mean?). If published, this will include your full peer review and any attached files.

Reviewer #1: **Yes: **Matias Arias-Aravena

Reviewer #2: No

<quillbot-extension-portal></quillbot-extension-portal>

---

## [Editor Report · Acceptance letter]

6 Jan 2023

PONE-D-22-27296R1 

Pathogenesis and defense mechanism while *Beauveria bassiana* JEF-410 infects poultry red mite, *Dermanyssus gallinae*

Dear Dr. Kim:

I'm pleased to inform you that your manuscript has been deemed suitable for publication in PLOS ONE. Congratulations! Your manuscript is now with our production department. 

Kind regards, 

on behalf of

Professor Shawky M Aboelhadid 

Academic Editor

PLOS ONE